# OpenReview forum: "DiFair: A Benchmark for Disentangled Assessment of Gender Knowledge and Bias"
_EMNLP/2023/Conference — EMNLP 2023 Findings_

### Official Review · Reviewer_118u · 2023-07-22

**Soundness:** 3

**Excitement:**

4: Strong: This paper deepens the understanding of some phenomenon or lowers the barriers to an existing research direction.

**Paper Topic And Main Contributions:**

This paper argues that previous benchmarks in model debiasing (about gender) fail to comprehensively evaluate the debiased models: While they involve gender-neutral situations, other situations, where gender knowledge is valuable and should be considered, are neglected. Therefore, the authors propose a human-annotated dataset named DIFAIR to solve the issue. This paper points out through experiments that previous debiasing methods actually achieve good debiasing performances at the cost of models’ gender knowledge. There remains a significant gap between the overall performance of models and that of humans. The authors’ additional findings also indicate the spurious correlations between gender and other factors like date.

**Questions For The Authors:**

Q1: Do you discover more spurious correlations between gender and other factors?
Q2: Since DIFAIR is proposed to evaluate the word representations, are the experimental results of model performances evaluated by DIFAIR consistent with that evaluated by the specific (downstream) tasks such as Stereoset?

**Reasons To Accept:**

This paper analyzed the gender knowledge reservation of debiasing models and constructed a new dataset to select those methods with both good gender-debiasing performance and rich gender knowledge. This is a good contribution toward the future development of fairness research in language models.

**Reasons To Reject:**

1. The performance of the recently proposed large language models is not evaluated. Lacking such experimental results is my primary concern. It is hard to estimate the contribution of DIFAIR to the model debiasing task: Whether DIFAIR can be a solved task? Is DIFAIR applicable to benchmarking LLMs?
2. The reproducibility of the paper may be a concern, as the authors did not mention whether the code or data will be publicly available for others to replicate or build upon this work.
3. The performance of more recent debiasing techniques such as Auto-Debias (guo et al., 2022) is missed in the experimental results, but it’s a minor issue.
Guo et al. Auto-Debias: Debiasing Masked Language Models with Automated Biased Prompts. ACL2022.

**Reproducibility:**

3: Could reproduce the results with some difficulty. The settings of parameters are underspecified or subjectively determined; the training/evaluation data are not widely available.

**Reviewer Confidence:**

4: Quite sure. I tried to check the important points carefully. It's unlikely, though conceivable, that I missed something that should affect my ratings.

---

> ### Author Rebuttal · Authors · 2023-08-29
>
> We thank the reviewer for their thorough review.
>
> ### Regarding the Raised Concerns:
>
> 1- As addressed in appendix A, line 853, DiFair dataset and all the code that is used to run the evaluations will be made publicly available after the publication of the paper and are currently available as supplementary material to this submission.
>
> 2- While DiFair is primarily designed for the evaluation of bi-directional models, it is indeed possible to apply it to recently developed autoregressive models through prompt engineering. This is acknowledged in section 8, line 639. Additionally, we have done a limited evaluation of DiFair on ChatGPT using 200 balanced instances of DiFair, which yielded a GIS of 71.48. The results demonstrate that, despite outperforming its smaller predecessors, the ChatGPT model still has a significant gap to human performance in terms of gender bias. We believe that the usage of DiFair to benchmark LLMs is an appropriate future work to bolster/refute our initial findings.
>
> 3- We thank the reviewer for bringing Auto-Debias into our attention. We were not able to evaluate all the prominent debiasing methods on our dataset due to resource and budget constraints. Given the constraints and the fact that many of the existing bias mitigation techniques have not released their pre-trained models, we have strived to make our experiments as inclusive as possible.
>
> During the rebuttal period, we managed to run additional experiments using Auto-Debias  on the BERT base model:
>
> | GSS | GNS | GIS |
> |-----|-----|-----|
> | 34.34 | 71.73 | 46.45 |
>
> These observations are consistent with the findings of the paper: debiased models tend to increase the GNS, while failing to retain useful gender information, thus decreasing the GSS (as a reminder, the original BERT base model has a GSS of 58.02, GNS of 63.91, and GIS of 60.82). Given that the same pattern is observed across five different debiasing techniques, one could expect to make similar observations on other techniques that are not specifically developed to retain useful gender information. This additional observation will be added to the camera-ready version.
>
> ### Regarding the Questions:
>
> 1- Our main focus in finding spurious correlations was the correlation between date and gender due to the availability of instances with the appropriate context. However, McMilin [1] have shown that such biases also exist for countries, and Subreddits. Given the consistency of our results with the finding of McMilin, we expect that GIS and DiFair are also able to extract such correlations by observing the change in score across a spectrum with respect to other variables, and can be used for the evaluation of techniques in removing spurious correlations. The relation between debiasing and spurious correlation is indeed a very interesting future research.
>
> 2- It's worth noting that diverse bias measurements often yield uncorrelated results, given their distinct definitions and quantification approaches. However, in our pursuit of comprehensively validating DiFair, we conducted an experiment specifically addressing the correlation between our metric and StereoSet. This choice is grounded in the fact that StereoSet's bias definition closely aligns with our work. The outcomes of this experiment are as follows:
> * GSS, LSM Correlation Coefficient of 0.7142 - GSS is most closely mapped to the Language Modeling Score in StereoSet.
> * GNS, SS Correlation Coefficient of 0.6517, GNS is most closely mapped to the Stereotype Score in StereoSet.
>
> We will add these additional results to the appendix of the camera-ready version of the paper.
>
> We again thank the reviewer for their constructive and thorough feedback.
>
> [1] Emily McMilin. 2022. Selection bias induced spurious correlations in large language models. In ICML 2022: Workshop on Spurious Correlations, Invariance and Stability.
>
> [2] Yang Trista Cao, Yada Pruksachatkun, Kai-Wei Chang, Rahul Gupta, Varun Kumar, Jwala Dhamala, and Aram Galstyan. 2022. On the Intrinsic and Extrinsic Fairness Evaluation Metrics for Contextualized Language Representations. In *Proceedings of the 60th Annual Meeting of the Association for Computational Linguistics (Volume 2: Short Papers)*, pages 561–570, Dublin, Ireland. Association for Computational Linguistics.

---

### Official Review · Reviewer_Mwxm · 2023-08-04

**Soundness:** 3

**Excitement:**

3: Ambivalent: It has merits (e.g., it reports state-of-the-art results, the idea is nice), but there are key weaknesses (e.g., it describes incremental work), and it can significantly benefit from another round of revision. However, I won't object to accepting it if my co-reviewers champion it.

**Missing References:**

See papers above.

**Paper Topic And Main Contributions:**

This paper introduces a new benchmark to study gender bias and
knowledge in masked language models. The samples in the benchmark
contain masked tokens that have to be filled with gender appropriate
nouns. Based on the preference of the model for gendered nouns, bias
scores are computed. The statements are constructed from Wikipedia and
Reddit. The samples are either gender neutral (the gender cannot be
determined by other words in the phrase) or gender-specific (one
specific gender is required to be filled for the masked token).


The bias scores are computed as the average absolute difference scores
for gendered tokens across samples in the benchmark. Given the two
types of sentences, the benchmark is able to assess both gender bias
(neutral sentences) and gender knowledge (gender specific sentences).

Several masked LMs are evaluated, alongside their distilled
counterparts. The experiments indicate that some improvement in
fairness may come at the cost of the degradation in gender
knowledge. The same seems to be true for a series of debiasing
techniques.


My main concern for this approach is in the evaluation. Intrinsic metrics for bias are shown (repeatedly) to be problematic for  measuring bias. In particular, I recommend the following studies:

https://aclanthology.org/2022.trustnlp-1.7.pdf
shows how simple rephrasing of sentences with different lexical choices but the same semantic meaning lead to widely different
scores

https://aclanthology.org/2021.acl-long.150.pdf
shows that intrinsic bias measures do not correlate with bias measured at the NLP task level

https://aclanthology.org/2022.naacl-main.122/
describes more issues related to bias metrics

https://aclanthology.org/2021.acl-long.81/
lists several issues with current datasets/benchmarks for bias auditing

I found the results on reduced gender knowledge with increased
fairness really interesting, but I'm not sure how to translate such an
experiment into a downstream task.

**Questions For The Authors:**

The bias score is not clear to me in the case of gender-specific
sentences. In the gender specific sentence, if the model prefers the
gender determined by the rest of the words, I don't think that's
bias. However, by the definition of the bias score, this may
happen. Could you please clarify? I think showing some examples with
the three different scores would be beneficial.

What is your take on intrinsic versus extrinsic measures of bias, especially in the light of the papers and results mentioned above?



**Reasons To Accept:**

New benchmark for gender bias
Interesting analysis between gender bias and gender knowledge

**Reasons To Reject:**

Intrinsic measures of bias do not correlate with extrinsic measures
and it is not clear how the models that are deemed more fair by
intrinsic models would behave in downstream tasks.

**Reproducibility:**

4: Could mostly reproduce the results, but there may be some variation because of sample variance or minor variations in their interpretation of the protocol or method.

**Reviewer Confidence:**

4: Quite sure. I tried to check the important points carefully. It's unlikely, though conceivable, that I missed something that should affect my ratings.

---

> ### Author Rebuttal · Authors · 2023-08-29
>
> ### Regarding the Raised Concern:
>
> We sincerely appreciate the thoughtful feedback provided by the reviewer, and we'd like to address their concerns while also providing a more comprehensive perspective on our work.
>
> It's crucial to acknowledge that intrinsic and extrinsic measures of bias have been extensively discussed in the literature. While the reviewer references Goldfarb-Tarrent et al.'s findings that intrinsic bias mitigation might not consistently translate to improved extrinsic bias metrics, it's important to note that the absence of a clear correlation doesn't inherently invalidate the significance of intrinsic measures. In this context, we want to highlight the work by Orgad et al. [1], who observed a similar lack of correlation in the reverse direction. This suggests that both intrinsic and extrinsic metrics offer distinct insights into bias behaviors within models. Our viewpoint is that intrinsic metrics help gauge the inherent bias encoded by models, while extrinsic metrics assess the propagation of this bias to downstream tasks.
>
> Moreover, the criticism towards bias metrics does not necessarily imply that intrinsic measures are less informative. Rather, it highlights the complexity of bias and its propagation. Our endeavor as researchers is to propose more nuanced and sophisticated metrics that mirror the intricacies of real-world tasks. Our metric, DiFair, is a step in this direction, employing a manually curated dataset and capturing the overall gender signaling capability of models, not just focusing on bias in isolation.
>
> ### Regarding the Question:
>
> Addressing your initial query regarding gender-specific sentences, we'd like to clarify that our intention is not to label models performing well in gender-specific settings as biased. Quite the opposite, we penalize models that demonstrate poor performance in gender-specific settings, even if they exhibit fairness in gender-neutral contexts. We invite you to refer to Section 2 of our work for a more detailed elucidation of our viewpoint on fairness, bias implications, and task formulation.
>
> To facilitate a clearer understanding of our approach, let's illustrate our calculation steps with an example using Figure 1. Imagine our gender-neutral sentence set contains a single sentence, "Since 2012, [MASK] has been a full professor." To achieve a GNS of 1, a model should minimize the distinction between masculine and feminine tokens when filling the MASK token, as indicated in the top example of Figure 1. Similarly, suppose that the gender-specific sentence set comprises a single sentence, “Achievements in the last tournament succeeded in bringing [MASK] back into the top 10 of BWF women's singles ranking.” A model aiming for a GSS of 1 should maximize the distance between predicted tokens based on the gender context of the MASK token. By applying the calculation steps delineated in Figure 1 using the probability distribution of tokens from a hypothetical model, a GSS of 0.9444 and a GNS of 0.9867 could be attained, ultimately resulting in a GIS of 0.9650. It's vital to comprehend that GSS and GNS calculations, while seemingly similar, operate on distinct data examples, rendering them separate evaluations. We encourage you to explore Appendix Section C, where you'll find a more detailed breakdown of sentences along with corresponding examples. This will provide you with a clearer insight into the distinction between gender-specific and gender-neutral sentences.
>
> In summary, we genuinely value the reviewer's input and their engagement with our work. We firmly believe that both intrinsic and extrinsic measures are indispensable in comprehending the multifaceted nature of bias in NLP models. DiFair, our proposed metric, contributes to this understanding by capturing more complex dimensions of model behavior, addressing some of the limitations present in existing evaluation methods. We hope that our work serves as a stepping stone toward more comprehensive and accurate evaluations of bias in language models.
>
> [1] Hadas Orgad, Seraphina Goldfarb-Tarrant, and Yonatan Belinkov. 2022. How Gender Debiasing Affects Internal Model Representations, and Why It Matters. In *Proceedings of the 2022 Conference of the North American Chapter of the Association for Computational Linguistics: Human Language Technologies*, pages 2602–2628, Seattle, United States. Association for Computational Linguistics.

---

### Official Review · Reviewer_kmvZ · 2023-08-05

**Typos Grammar Style And Presentation Improvements:** 109
**Soundness:** 3

**Excitement:**

4: Strong: This paper deepens the understanding of some phenomenon or lowers the barriers to an existing research direction.

**Paper Topic And Main Contributions:**

The paper contributes two portions:1. Dataset creation and 2.Evaluation and comparison empirically with existing standards.
The author contributed well for the 1.Data set -DIFAIR- creation part with data collection, data cleaning, data labeling using annotators, annotation guidelines, revaluation of data set qualify is described well in paper.

Proposed new bias evaluation metric -GIS - specifically for measuring gender bias in Transformer-based MLM model. This metric is used to evaluate the various transformer based models using DIFAIR dataset for bias (gender) detection/fairness by the models.
The author claims that the metric can be used to measure the bias detection and how much the model is fair in retaining the gender knowledge. The results using, GSS, GNS, GIS score for DIFAIR shows that some models are better in bias reduction but weak in gender information, some retains gender info but weak in bias reduction.

The proposed DIFAIR is not empirically compared with the most relevant dataset CrowS-Pairs and StereoSet.

The proposed GIS metric is not empirically compared with other bias evaluation metric for gender.

Data visualization of vector embeddings may help to show that gender knowledge is retained or fairness of model.

**Questions For The Authors:**

Write mathematical equations separately for GNS, GIS

112,115: models that exhibit representation harm, allocational harm ....---> references?

Why not compared your metric with other metric? Or why not used your GIS metric for other dataset and show that gender info preservation can be measured?

Apart from your results using GSS, GNS, GIS for DIFAIR, any other empirical study / evaluation or mathematical proof to show that gender information preservation by model can be measured by GIS?
The visualization of embeddings - vector representation of model may be used to show that the gender knowledge is retained by the model.



**Reasons To Accept:**

1.Proposed DIFAIR manual curated dataset for evaluating Transformer based MLM model for gender bias.

2. Proposed a new Metric GIS to measure gender bias and check the gender knowledge is retained by the model /fair enough in preserving the gender information for required sentences.

**Reasons To Reject:**

No reason

**Reproducibility:**

4: Could mostly reproduce the results, but there may be some variation because of sample variance or minor variations in their interpretation of the protocol or method.

**Reviewer Confidence:**

4: Quite sure. I tried to check the important points carefully. It's unlikely, though conceivable, that I missed something that should affect my ratings.

---

> ### Author Rebuttal · Authors · 2023-08-29
>
> We thank the reviewer for their thorough review and helpful comments.
>
>
> ### Regarding the Questions:
>
> 1- Regarding the reference for representational and allocative harms, we will add an appropriate citation in the camera-ready version. It was first mentioned in the context of Machine Learning by Kate Crawford in a Keynote at NIPS 2017 [1].
>
> 2- It's important to recognize that not all intrinsic bias measurements are directly comparable due to their varying definitions of bias behavior and the diverse methodologies they employ to quantify bias. This discrepancy arises from the complexity of the task itself. For instance, in the context of our DiFair metric, bias is evaluated by a model's ability to maintain fairness in gender-neutral contexts while also retaining valuable gender information for gender-specific contexts. In contrast, metrics like CrowS-Pairs define gender bias as the proportion of stereotypical behavior relative to anti-stereotypical behavior.
> Rather, it's more appropriate to anticipate consistent patterns across diverse metrics. Notably, our findings align with metrics such as StereoSet and CrowS-Pairs regarding the effect of model size. Our work also shares congruence with the studies of Delobelle and Berendt, who employ LPBS and DisCo to demonstrate reduced bias in distilled models, and McMilin, who employs selection bias criteria to uncover spurious correlations involving gender in Wikipedia data.
>
> We'd also like to direct your attention to our response to *Reviewer 118u*, where we delve into the correlation evaluation between our scores and StereoSet scores. This is particularly relevant as StereoSet aligns closely with our study's bias definition.
>
> Regarding the application of GIS to other bias datasets, it's crucial to consider the practical limitations. Calculating GIS necessitates additional annotation efforts, including the labeling of sentences as gender-neutral or gender-specific, and the replacing of words like names and pronouns with special tokens. This annotation process is both resource-intensive and impractical to carry out at scale. For a deeper understanding of the intricacies involved, we invite you to explore Section 3 and Appendix Section D.2 of our work. These sections provide comprehensive insights into the complexities of the annotation process.
>
> 3- Further work is indeed required to show that GIS fully captures the amount of gender information preservation, and it is a part of a future work to show that gender preservation metrics conform with GIS. Currently, our work is consistent with Orgad et al. [2], who, as a part of their work, show that Oversampling and Undersampling consistently decrease the gender information compression, a preservation metric, albeit in a more extrinsic setting. This is consistent with our results, and is observable in the CDA section of Table 3. Additionally, Debiasing methods consistently decrease GSS, while increasing GNS. Given that GNS is evaluated as the model’s ability to indiscriminately choose a gender, while GSS is evaluated as the model’s ability to retain the gender when required, we believe that this is a good indicator that GIS indeed captures the amount of gender information preservation.
> Furthermore, we welcome the reviewer’s suggestion of visualizing the vector embeddings. However, we believe that visualization of contextual embeddings with respect to a single dimension (gender) is especially challenging without the introduction of new metrics. We would welcome any suggestions in this regard, and nonetheless will look for methods to apply the visualization. To better showcase the effectiveness of GSS in capturing gender information, we will add additional examples to the appendix in the camera ready version.
>
> ### Regarding the Typos and Presentation Improvements:
>
> We will take into account all the typos and formatting concerns (presentation of the formulas, wording) and address them in the camera-ready version of the paper. We thank the reviewer for their attention to details.
>
> We again thank the reviewer for their constructive and thorough feedback.
>
> [1] Crawford, K., 2017. Conference on Neural Information Processing Systems, invited speaker.
>
> [2] Hadas Orgad, Seraphina Goldfarb-Tarrant, and Yonatan Belinkov. 2022. How Gender Debiasing Affects Internal Model Representations, and Why It Matters. In *Proceedings of the 2022 Conference of the North American Chapter of the Association for Computational Linguistics: Human Language Technologies*, pages 2602–2628, Seattle, United States. Association for Computational Linguistics.

---

### Meta-Review · Area_Chair_yAxy · 2023-09-17

**Recommendation:** 4

**Metareview:**

The paper introduces "DIFAIR", a manually curated dataset designed to evaluate Transformer-based Masked Language Models (MLM) for gender bias. The dataset's creation involved data collection, cleaning, labeling using annotators, and establishing annotation guidelines. A new metric, GIS, is proposed to measure gender bias and assess whether a model retains gender knowledge or is fair in preserving gender information in relevant sentences. The paper highlights that while some models reduce bias, they may compromise gender knowledge, and vice versa. The research emphasizes the need for a balance between reducing bias and retaining essential gender information. However, the paper does not compare DIFAIR with other relevant datasets like CrowS-Pairs and StereoSet, nor does it compare the GIS metric with other gender bias evaluation metrics.

The paper's significant contributions include the introduction of the DIFAIR dataset, specifically designed to evaluate Transformer-based MLMs for gender bias. The proposed GIS metric offers a novel way to measure gender bias while ensuring that models retain essential gender knowledge. The paper's analysis provides valuable insights into the trade-off between reducing gender bias and preserving gender knowledge in models. The research addresses a crucial gap in the current understanding of model debiasing, emphasizing the importance of gender knowledge retention. Overall, the paper's contributions are seen as valuable for advancing fairness research in language models.

Some weaknesses, listed below, have been discussed during the rebuttal phase.
The research does not evaluate the performance of recently proposed large language models, leaving a gap in understanding the dataset's applicability and relevance to state-of-the-art models. The paper's reproducibility is questionable, as there's no mention of making the code or data publicly available, which is essential for the broader research community. Additionally, the paper misses out on evaluating the performance of recent debiasing techniques.

---

### Decision · Program_Chairs · 2023-10-07

**Decision:**

Accept-Findings

**Comment:**

The paper introduces "DIFAIR", a manually curated dataset designed to evaluate Transformer-based Masked Language Models (MLM) for gender bias. The dataset's creation involved data collection, cleaning, labeling using annotators, and establishing annotation guidelines. A new metric, GIS, is proposed to measure gender bias and assess whether a model retains gender knowledge or is fair in preserving gender information in relevant sentences. The paper highlights that while some models reduce bias, they may compromise gender knowledge, and vice versa. The research emphasizes the need for a balance between reducing bias and retaining essential gender information. However, the paper does not compare DIFAIR with other relevant datasets like CrowS-Pairs and StereoSet, nor does it compare the GIS metric with other gender bias evaluation metrics.

The paper's significant contributions include the introduction of the DIFAIR dataset, specifically designed to evaluate Transformer-based MLMs for gender bias. The proposed GIS metric offers a novel way to measure gender bias while ensuring that models retain essential gender knowledge. The paper's analysis provides valuable insights into the trade-off between reducing gender bias and preserving gender knowledge in models. The research addresses a crucial gap in the current understanding of model debiasing, emphasizing the importance of gender knowledge retention. Overall, the paper's contributions are seen as valuable for advancing fairness research in language models.

Some weaknesses, listed below, have been discussed during the rebuttal phase.
The research does not evaluate the performance of recently proposed large language models, leaving a gap in understanding the dataset's applicability and relevance to state-of-the-art models. The paper's reproducibility is questionable, as there's no mention of making the code or data publicly available, which is essential for the broader research community. Additionally, the paper misses out on evaluating the performance of recent debiasing techniques.